

# The role of monocyte chemoattractant protein-1 (MCP-1) as an immunological marker for patients with leprosy: a systematic literature review

Flora Ramona Sigit Prakoeswa[1,2], Ellen Josephine Handoko[3], Erika Diana Risanti[4], Nabila Haningtyas[3], Nasrurrofiq Risvana Bayu Pambudi[5] and Muhana Fawwazy Ilyas[3]

[1] Department of Dermatology and Venereology, Faculty of Medicine, Muhammadiyah University of Surakarta, Surakarta, Central Java, Indonesia
[2] Department of Dermatology and Venereology, PKU Muhammadiyah Surakarta Hospital, Surakarta, Central Java, Indonesia
[3] Faculty of Medicine, Sebelas Maret University, Surakarta, Central Java, Indonesia
[4] Department of Biomedical, Faculty of Medicine, Muhammadiyah University of Surakarta, Surakarta, Central Java, Indonesia
[5] Faculty of Medicine, Muhammadiyah University of Surakarta, Surakarta, Central Java, Indonesia

## ABSTRACT

Leprosy, a significant global health concern affecting primarily the peripheral nerves and integumentary system, is influenced by the host immune system's response, affecting its pathology, disease progression, and reaction occurrence. MCP-1, integral to leprosy's immunological processes, holds promise as a diagnostic tool and predictor of reaction occurrence. This systematic review aimed to investigate MCP-1's involvement in leprosy. Literature search, employing specified MeSH keywords, covered databases such as PubMed, Scopus, ScienceDirect, and Wiley Online Library until September 30th, 2023, yielding seventeen relevant studies. Assessing each study's quality with the Newcastle-Ottawa Scale (NOS) and investigating bias using the Risk of Bias Assessment tool for Non-randomized Studies (RoBANS), a narrative synthesis compiled findings. Seventeen distinct studies were included, each characterized by diverse designs, sample sizes, demographics, and outcome measures, highlighting MCP-1's potential in diagnosing leprosy, differentiating it from control groups, and discerning leprosy types. Furthermore, MCP-1 shows promise in predicting leprosy reversal reactions. Although MCP-1 offers clinical benefits, including early diagnosis and type differentiation, further research with larger sample sizes and standardized methodologies is imperative to confirm its diagnostic utility fully.

## INTRODUCTION

Leprosy first appeared in an Egyptian skeleton from the second century BCE and was first documented in India (600 BCE), therefore is one of the earliest diseases that afflict mankind (*De Souza et al., 2016*; *Prakoeswa et al., 2021a*). *Mycobacterium leprae* is the

Corresponding author
Flora Ramona Sigit Prakoeswa,
frsp291@ums.ac.id

infectious agent that causes leprosy, a chronic granulomatous disease primarily affecting the integumentary system and peripheral Click or tap here to enter text.nerves (*Medeiros et al., 2015*). By changing the mitochondrial glucose metabolism in Schwann cells (SC), *M. leprae* infects both macrophages and these cells (*Angst et al., 2020*). WHO data from 2021 demonstrate that there are 133.781 cases and 140.546 new cases, with India, Brazil, and Indonesia continuing to contribute a significant number of new cases of leprosy worldwide (74%) (*Ariyanta & Muhlisin, 2017*). In 2018, 17,439 new cases of leprosy were reported in Indonesia, 1,121 of which had grade-2 disabilities (G2D) (*OMS, 2019*; *Prakoeswa et al., 2021b*; *Prakoeswa et al., 2021c*).

The susceptibility of an individual to leprosy is established by multiple variables: idiosyncratic, immunological, and environmental factors of the host (*Sauer et al., 2015*). Transmission by upper respiratory secretions can occur from prolonged interaction with untreated leprosy patients (*Alinda et al., 2020*; *Maymone et al., 2020*; *Prakoeswa et al., 2024*). Symptoms might vary from person to person due to immunogenic differences that result in a particular clinical appearance (*Froes, Trindade & Sotto, 2022*). Clinical diagnosis of leprosy is confirmed if one out of three cardinal signs are present: Cutaneous lesions, such as macules or plaques with hypopigmentation or erythema, accompanied with the loss of sensation on the skin; Thickening or enlargement of peripheral nerves and signs of its damage, such as loss of sensory, paralysis or motoric dysfunction with or without nerve enlargement; Findings of acid-fast bacilli (AFB) on skin biopsy and/or lesion scraping (*Griffiths et al., 2016*).

*Mycobacterium sp.* is one of the acid-fast bacilli due to their capacity to withstand acid-induced color loss during staining processes (*Reynolds, Moyes & Breakwell, 2009*; *Prakoeswa, Rumondor & Prakoeswa, 2022*). *M. leprae* has a highly specific antigen which is phenolic glycolipid-I (PGL-I) and it has the ability to attach to the basal lamina of Schwann cell-axon units (*Penna et al., 2016*; *Gautam et al., 2021*). Toll-like receptors (TLR) identify PGL-I and present it to APC. APC introduces *M. leprae* to lymphoid naïve T-cells which then can transform into Th1, Th2, Treg, and Th17 (*Prakoeswa et al., 2020b*). Leprosy develops because of an imbalanced immune response, marked by T-cell dysfunction, heightened cell death, and an imbalance between the Th1 and Th2 immune responses (*Endaryanto, Ramona Sigit Prakoeswa & Rosita Sigit Prakoeswa, 2020*). Th1 dominant immune responses are mediated by protective IFN-$\gamma$ and IL 2 with microbicidal properties which is more prevalent in PB type leprosy (*Richardus et al., 2018*; *Prakoeswa et al., 2020a*; *Prakoeswa et al., 2020b*). MCP-1 is associated with Th1 responses and has an antagonistic association with IFN-$\gamma$, which both cytokines play a crucial role in *M. leprae* elimination (*Endaryanto, Ramona Sigit Prakoeswa & Rosita Sigit Prakoeswa, 2020*; *Queiroz et al., 2021*). In addition, it is well recognized that the family of transcription factors named nuclear factor kappa B (NF $\kappa$B) plays a central role in the modulation of innate and adaptive immunity (*Wambier et al., 2014*; *Hadi et al., 2021*).

Chemotactic cytokines are classified into two main classes (CXC and CC) and manage how other cells response to a chemical stimulation (chemotaxis) (*Bikfalvi & Billottet, 2020*). Monocyte chemoattractant/chemotactic protein (MCP-1)/CC chemokine ligand-2 (CCL2), a member of the CC chemokine family, is involved in regulation of

monocyte, microglia, and memory T cell passage and penetration to the site of injury and infection in a variety of diseases (*Kumar et al., 2016*; *Singh, Anshita & Ravichandiran, 2021*). MCP-1 has been identified as a potent inducer of macrophage infiltration, a reliable marker of inflammation, and a potential therapeutic target for a variety of inflammatory illnesses (*Geluk, 2013*). Since MCP-1 facilitates the recruitment of macrophages to the leprosy nerves, it is possible that MCP-1 is related to the severe nerve fibrosis (*Medeiros et al., 2015*). MCP-1 is significantly higher in PB patients, however some literatures stated MCP-1 is higher in MB patients (*Chen et al., 2019*; *Gautam et al., 2021*; *Yuan et al., 2021*). MCP-1 indicates a more vigorous reaction to *M. leprae* (*De Carvalho et al., 2017*).

MCP-1 is useful in understanding the pathogenesis of leprosy because of its involvement between *M. leprae* and host cells' immune system (*Hirai et al., 2018*). MCP-1 can be used to determine the degree of inflammation in a variety of medical conditions (*Singh, Anshita & Ravichandiran, 2021*). Due to the difference in expression between PB/MB and TT/BT leprosy patients, MCP-1 could be utilized to distinguish between different types of leprosy (*Chen et al., 2019*; *Gautam et al., 2021*; *Yuan et al., 2021*). MCP-1 was found sensitive only to PB leprosy. MCP-1 can also be used as an additional marker to enhance the accuracy of leprosy diagnosis because the current diagnostic testing for IgM antibodies against PGL-I is not able to represent household leprosy contacts (*Silva et al., 2023*). With IFN-$\gamma$, MCP-1 are potential indicators of subclinical infection of *M. leprae* in household contacts, also as a parameter of early infection monitoring (*Queiroz et al., 2021*). MCP-1 is currently under investigation as a potential immunotherapy as shown in previous study which immunotherapy with *Mycobacterium* vaccine has shown benefit to MB leprosy patients (*Geluk, 2013*; *Pandhi & Chhabra, 2013*). Therefore, the goal of this systematic review is to completely synthesize all findings on MCP-1's potential as a biomarker to diagnose and distinguish different types of leprosy, as well as its potential as a therapeutic intervention.

## Survey methodology
### Study design

The review protocol for this investigation was registered with the International Prospective Register of Systematic Reviews (PROSPERO; ID: CRD42023460380), and the study was conducted in accordance with the Preferred Reporting Items for Systematic Reviews and Meta-Analyses (PRISMA). The search was conducted in October 2023 in four databases (PubMed, Scopus, Wiley Online Library, ScienceDirect). Medical subject headings (MeSH)-based keywords were utilized in the search approach. Keywords used were: (''Leprosy'' OR ''Hansen disease'' OR ''Hansen's disease'' OR ''Morbus Hansen'' OR ''Leprae'') AND (MCP-1 OR CCL2 OR CCL#2 OR ''Chemokine (C-C Motif) Ligand#2'' OR ''C-C chemokine ligand#2'' OR MCP1 OR MCP#1 OR ''Monocyte Chemotactic and Activating Factor'' OR ''Monocyte Chemoattractant Protein#1'' OR ''Monocyte Chemotactic Protein# 1'') AND (''Immunological Marker'' OR ''Immunologic Marker'' OR ''Immunological Marker$'' OR ''Marker$'' OR ''Biomarker$''). We used these keywords for PubMed, Scopus, and Wiley Online Library. For ScienceDirect, we use: (''Leprosy'' OR ''Hansen's disease'' OR ''Leprae'') AND (''MCP-1'' OR ''CCL2'' OR ''CCL#2'') AND (''Immunological

Marker$'' OR "Biomarker$''). To ensure that no pertinent papers were overlooked, reference lists of the included studies were reviewed.

### Inclusion and exclusion criteria

Studies giving data regarding leprosy, MCP-1, and immunological markers up until September 30th, 2023, were evaluated. Only observational studies in humans were included. However, we only included publications that were written in English. All types of reviews are excluded from this study. No time limitations were placed on this study.

### Study selection

The screening process began by importing all search results upon titles and abstracts into rayyan.ai, and duplicate articles were subsequently excluded. F.R.S.P. and E.J.H, two reviewers, independently examined the obtained articles' titles, abstracts, and full texts in accordance with inclusion and exclusion criteria. A third reviewer (E.D.R) arbitrated any disagreements between the two reviewers.

### Quality assessment

To assess each quality of the study, we used Newcastle-Ottawa Scale (NOS) that consisted of three major items: selection of study groups (0–4 points), comparability of cases and control studies (0–2 points) or cohorts, and ascertainment of exposure/outcome (0–3 points). This scale applies to cohort and case control study, however for cross-sectional study, the NOS items were selection of study group (0–5 points), comparability of cases and control studies (0–1 points) and ascertainment of exposure/outcome (0–3 points). Studies were considered high-quality if they received six points or higher. This assessment of study quality was conducted by two reviewers, F.R.S.P and E.J.H, with any disparities resolved through the intervention of a third reviewer, E.D.R.

### Risk of bias assessment

The Risk of Bias assessment tool for non-randomized research (RoBANS) is utilized to evaluate the potential for bias in the research that are incorporated. This tool consists of six items: participant selection, confounding variables, exposure measurement, blinding of outcome assessments, incomplete outcome data, and selective outcome reporting. Two independent reviewers, namely F.R.S.P and E.J.H, carried out the risk of bias assessment using RoBANS. In the event of any discrepancies or disagreements, a third reviewer (E.D.R) was consulted to reach a consensus.

### Data analysis

Information such as the country of testing, study design, MCP-1 measurement, and time of MCP-1 measurement were all gathered from previous studies and reviewed by E.J.H, N.H. and M.F.I. A qualitative analysis was then conducted to cross-examine all the findings.

## RESULTS

### Study selection

This systematic review was carried out using the Preferred Reporting Items for Systematic Reviews and Meta-Analyses (PRISMA) guideline. We retrieved a total of 97 studies from
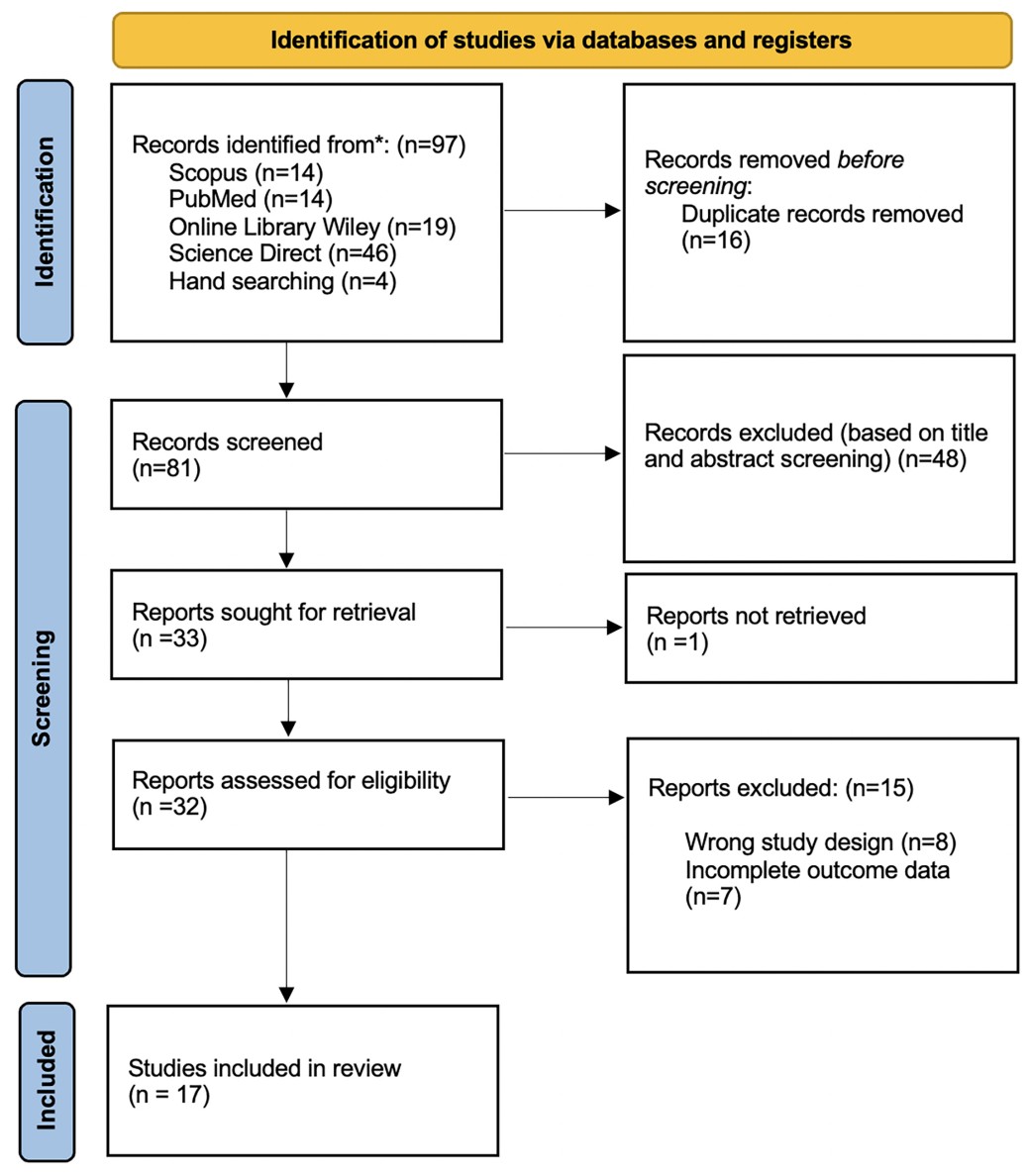

**Figure 1  PRISMA 2020 flow diagram.**

the following databases: Scopus ($n = 14$), PubMed ($n = 14$), Wiley Online Library ($n = 19$), ScienceDirect ($n = 46$), and hand searching ($n = 4$). We eliminated 16 duplicate studies before commencing the screening process. Following a review of titles and abstracts, we excluded 48 studies. Unfortunately, one article could not be retrieved. The remaining thirty-two articles were assessed for eligibility; eight studies were eliminated due to an inaccurate study design, and seven studies were removed due to insufficient outcome data. Finally, seventeen studies were included in the review. All review processes are described in Fig. 1.

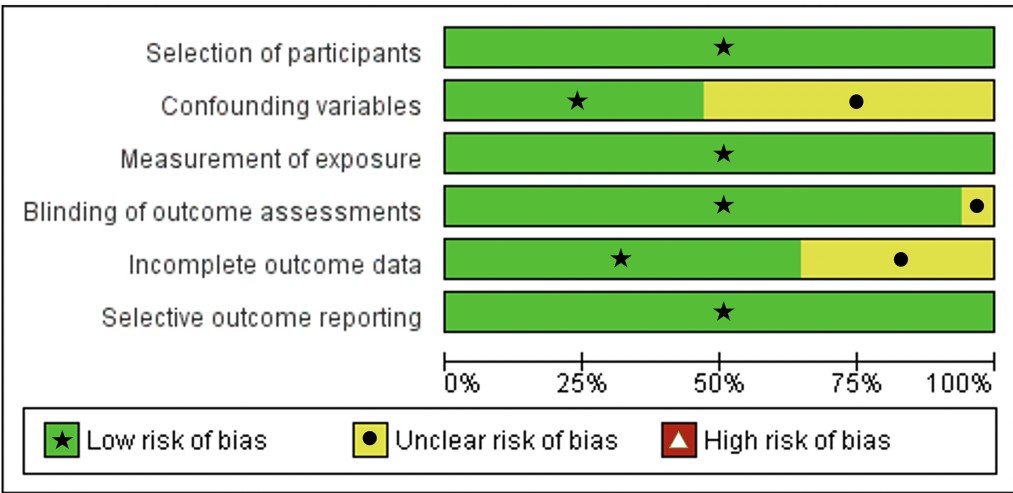

**Figure 2  Risk of Bias Assessment tool for non-randomized studies (RoBANS) graph.**

## Quality and risk of bias

Our studies' eligibility was assessed further for its quality using Newcastle-Ottawa scale (NOS) instrument and risk of bias using Risk of Bias Assessment tool for non-randomized Studies (RoBANS) tool. The results of the quality assessment were presented in Table 1. All the included studies scored at least six points in the quality assessment. Following, risk of bias assessment results using the RoBANS tool were presented in Figs. 2 and 3, in which most of the items have 'low' scores. However, some confounding variables items and incomplete outcome data were 'unclear.' We marked these items unclear because there was not any information in the passage explaining those items.

## Study characteristics

Majority of study designs were cross-sectional ($n = 8$), others were case controlled studies ($n = 4$) and cohort studies ($n = 5$). Samples were varied from 8 to 1,332, with a total sample size of 2121 patients. Studies varied from multiple countries. Comprehensive explanation of study characteristics can be observed in Table 1.

## Study results
## MCP-1 as a potential diagnostic marker in leprosy

From our systematic review, we found that MCP-1 has potential diagnostic abilities (*Geluk et al., 2012*; *Meneses et al., 2014*; *Medeiros et al., 2015*). A cross-sectional study by *Medeiros et al. (2015)* in 23 Pure Neural Leprosy (PNL) patients found MCP-1' immunoreactivity in PNL Schwann cells' biopsy samples from either Acid-Fast Bacilli (AFB)$^+$ or AFB$^-$. MCP-1 was detected in 13 out of 23 PNL patients (66.7% in PNL AFB$^+$ & 81.8% in PNL AFB$^-$). MCP-1 expression showed a correlation with fibrosis that was not influenced by HLA-DR, CD3, CD4, CD8, CD45RA, CD68, or any other immunologic markers ($p = 0.026$) (*Medeiros et al., 2015*). A global gene expression profile of *Mycobacterium leprae*-infected primary human Schwann cells identified the genes differentially expressed in the type 1

**Table 1  Characteristic of study.**

| No. | Author (year) | Country | Study design | MCP-1/CCL2 measurement | Time of MCP-1 measurement | Newcastle-Ottawa scale (NOS) score |
|---|---|---|---|---|---|---|
| 1 | *Mendonça et al. (2010)* | Brazil | Cross-sectional | Chemokine concentrations were assessed in blood samples using sandwich ELISA kits that included CCL2, CCL3, CCL11, and CCL24 kits. | At admission | 7 |
| 2 | *Geluk et al. (2010)* | Netherlands | Cross-sectional | Whole blood was drawn and after incubation in a 48-well plate with antigen, flow cytometry was performed. Then determination of cytokines and chemokines, including CCL2 was performed. | At admission | 7 |
| 3 | *Meneses et al. (2014)* | Brazil | Cross-sectional | The sandwich enzyme-linked assay (ELISA) was used to measure urinary MCP-1. | Approximately after an 8-hour fasting period. | 7 |
| 4 | *Medeiros et al. (2015)* | Brazil | Cross-sectional | Anti-MCP1 mouse monoclonal antibodies were used to stain MCP-1 then observed under a microscope. | Upon admission, following a 6-month multidrug regimen for PB leprosy and a 12-month regimen for MB leprosy. | 8 |
| 5 | *De Toledo-Pinto et al. (2016)* | Brazil | Cross-sectional | MCP-1 was measured using a multiplex biometric immunoassay. | At admission | 8 |
| 6 | *Angst et al. (2020)* | Brazil | Cross-sectional | Sensory nerve was biopsied, then serum cytokine levels, histopathological evaluation, clinical and neurophysiological evaluations were done. | At admission | 8 |
| 7 | *Dias et al. (2021)* | Brazil | Cross-sectional | In culture supernatants from A549 cells, MCP-1 concentration was assessed by ELISA. | At admission | 7 |
| 8 | *Cunha et al. (2023)* | Brazil | Cross-sectional | Whole blood was extracted and utilized for TLR4 genotyping with PCR and chemokine and cytokine measurement using a cytometric beads array. | At admission | 7 |
**Table 1** (*continued*)

| No. | Author (year) | Country | Study design | MCP-1/CCL2 measurement | Time of MCP-1 measurement | Newcastle-Ottawa scale (NOS) score |
|-----|---------------|---------|--------------|------------------------|---------------------------|-----------------------------------|
| 9 | *Stefani et al. (2009)* | Brazil | Case control | Plasma aliquots were kept at −80 °C after blood was taken in EDTA and centrifuged then frozen at −80 °C. After being thawed and centrifuged at 1,000 × g for 10 min at 4 °C, the supernatant was filtered and used shortly thereafter. | At admission | 7 |
| 10 | *Bobosha et al. (2012)* | Brazil, Ethiopia, and The Netherlands | Case Control | MCP-1 was measured using multiplex technology and analysed with Bio-Plex Manager Software 4.0. | At admission (before the initiation of MDT). | 7 |
| 11 | *Santana et al. (2017)* | Brazil | Case control | Whole blood was collected by venepuncture and centrifuged at 20,000g for 10 min to obtain serum. Then, ELISA chemokine assays were performed to measure levels of MCP-1. | At admission | 8 |
| 12 | *Biswas et al. (2024)* | India | Case control | Detection of MCP-1 was performed using polymerase chain reaction-restriction fragment length polymorphism (PCR-RFLP). | At admission | 7 |
| 13 | *Geluk et al. (2012)* | Bangladesh, Brazil, Ethiopia, South Korea | Cohort | Whole blood was drawn and after incubation in a 48-well plate with antigen, flow cytometry was performed. Then determination of cytokines and chemokines, including CCL2 was performed. | Approximately 24 h after whole blood was drawn. | 6 |
| 14 | *De Carvalho et al. 2017* | Brazil | Prospective cohort | MCP-1 was measured using a multiplex biometric immunoassay. | T0 (0-3 months before BCG vaccination); T1 (6-26 months from onset of their index case treatment). | 8 |
| 15 | *Tió-Coma et al. (2019)* | Bangladesh, Brazil, Ethiopia, Nepal, Netherlands | Prospective cohort | Following the process of RNA separation, dual color reverse-transcription multiplex ligation-dependent probe amplification (dcRT-MLPA) tests were run. The result then analysed and transcriptomic risk factors was identified. | At admission (before the initiation of MDT). | 7 |

**Table 1** (*continued*)

| No. | Author (year) | Country | Study design | MCP-1/CCL2 measurement | Time of MCP-1 measurement | Newcastle-Ottawa scale (NOS) score |
|---|---|---|---|---|---|---|
| 16 | *Queiroz et al. (2021)* | Brazil | Cohort | Serum levels of chemokines CXCL8 (IL-8), CCL2, CXCL9, and CXCL10, as well as cytokines TNF, IL-6, IFN-$\gamma$, IL-2, IL-17A, IL-4, and IL-10, were tested in all patients after blood was obtained. | 2014 (Time 0-T0) and 2015 (Time 1-T1). | 8 |
| 17 | *Yuan et al. (2021)* | China | Cohort | RNA sequencing was performed using The DESeq algorithm from peripheral blood that was centrifuged. Data was confirmed using RT-qPCR. | 12 h after blood was drawn | 7 |

IFN pathway, among them, the gene encoding $2'$–$5'$ oligoadenylate synthetase-like (OASL) underwent the greatest upregulation in *M. leprae* infected human macrophage. OASL inactivate modulated *M. leprae*-triggered MCP-1 induction (*De Toledo-Pinto et al., 2016*). A cohort study conducted on 160 patients by *Geluk et al. (2012)* discovered that MCP-1 (or CCL2) was considerably increased in TT/BT patients following stimulation with *M. leprae* in contrast to endemic controls (ECs) ($p = 0.0021$). In Bangladesh, there was good to excellent differentiation between the TT/BT and EC groups, as indicated by the MCP-1 area under the curve (AUC) of 0.94 (*Geluk et al., 2012*). This finding was supported by case control study conducted by *Bobosha et al. (2012)* that after administration of ML1601c peptide from *Mycobacterium leprae*, healthy endemic controls has significantly higher levels of MCP-1 ($p = 0.0347$) (*Bobosha et al., 2012*). BCG vaccination increases *in vitro* levels of pro-inflammatory mediators was observed in household contacts of multibacillary leprosy patients (HCMB) after BCG vaccination especially IL-1 $\beta$, IL-6, MCP-1 and MIP-1 $\beta$ (*De Carvalho et al., 2017*). Study conducted by *Meneses et al. (2014)* on 44 patients found that leprosy patients had higher urinary MCP-1 (101.0 $\pm$79.8 *vs.* 34.5 $\pm$14.9 mg/g-Cr, $p = 0.006$) and urinary MDA levels (1.77 $\pm$1.31 *vs.* 1.27 $\pm$0.66 mmol/g-Cr, $p = 0.0372$) than healthy controls (*Meneses et al., 2014*). A similar finding was found in a case-control conducted by *Biswas et al. (2024)* that MCP-1 level was significantly higher in all leprosy patients compared to healthy controls ($p = 0.0001$) (*Biswas et al., 2024*).

One cross-sectional study (*Dias et al., 2021*) discovered MCP-1 response in activated and inactivated *M. leprae* in A59 alveolar epithelial cells. 24 h of incubation resulted in higher MCP-1 levels ($p < 0.05$) in the treated cells' supernatants compared to control cells. At a later stage of incubation (48 h), only bacteria that had been inactivated could cause MCP-1 to be produced ($p < 0.05$). The impact of the pharmacological inhibitor wedelolactone on MCP-1 was also investigated in this research, but no effect was found; hence, its regulation is controlled by a different mechanism that is independent of NF-kB (*Dias et al., 2021*). Several studies were investigating MCP-1 genetic properties (*Santana et al., 2017*; *Cunha et al., 2023*). Carriers of the TT genotype (TC and TT) in TLR1_rs5743551

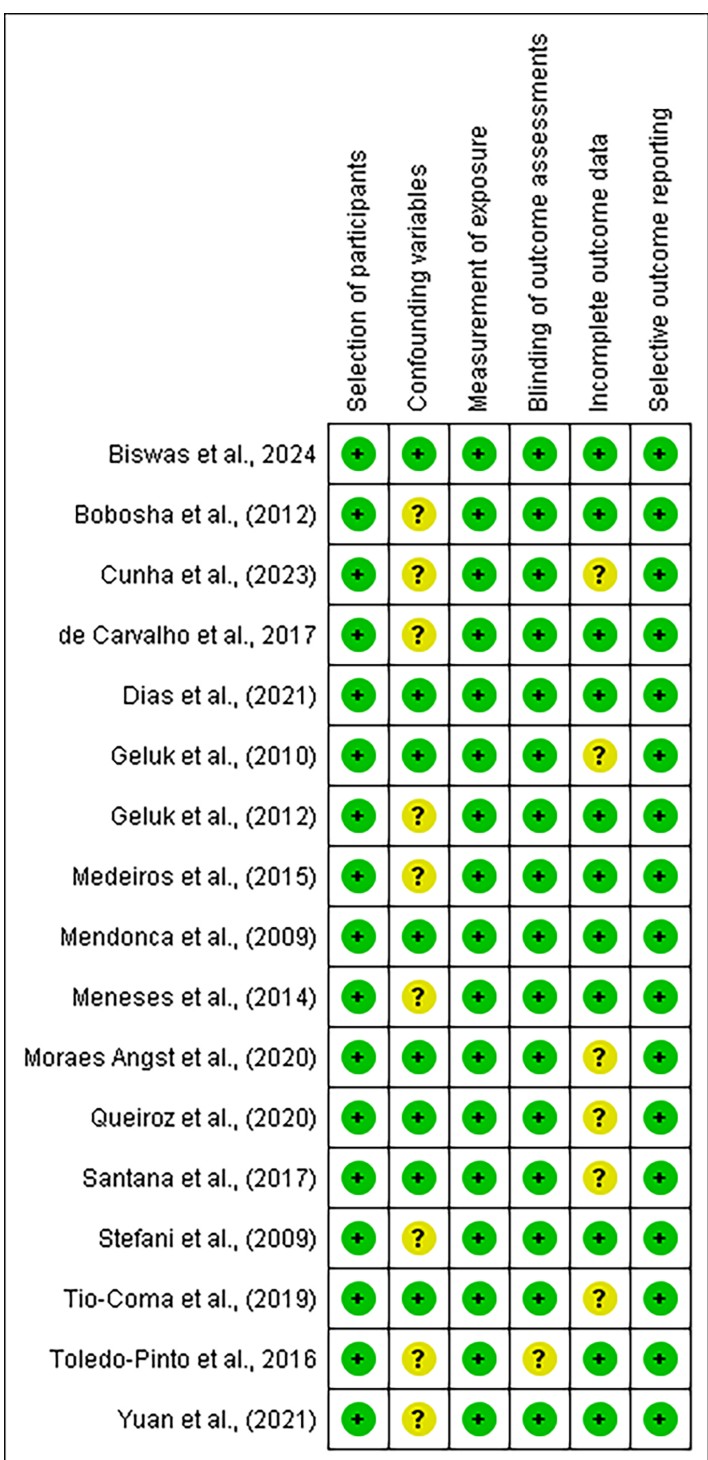

**Figure 3** **Summary of Risk of Bias Assessment tool for non-randomized studies (RoBANS).**

produced decreased serum levels of MCP-1, according to *Santana et al. (2017)*. Based on TLR4 rs1927914 alleles/genotype, *Cunha et al. (2023)* found that the AA genotype (CXCL8, MCP-1, TNF, and IL-2) was linked to a more prominent secretion *in vitro* culture of HHC (PB) and HHC (MB) (*Cunha et al., 2023*). *Biswas et al. (2024)* also stated AA, AG and GG genotypes of CCL2-2518 A > G SNP ($p = 0.0001$) (*Biswas et al., 2024*). Another study compared the levels of MCP-1 between leprosy neuropathy and diabetic neuropathy. This was performed by a cross-sectional study in Brazil by *Angst et al. (2020)* that found MCP-1 value in diabetic neuropathy group was statistically significant compared to leprosy neuropathy group ($p = 0,001$ and $p = 0,01$) (*Angst et al., 2020*).

### Role of MCP-1 in leprosy diagnosis, subtyping, and reversal reaction prediction

MCP-1 can also be used in discriminating types of leprosy (*Tió-Coma et al., 2019*; *Yuan et al., 2021*). A cohort study by *Yuan et al. (2021)* on 82 patients found that MCP-1 showed an excellent performance in diagnosing types of leprosy. Between leprosy patients *vs.* endemic controls (ECs) with AUC of 0.87 (95% CI [0.75–0.98]), sensitivity of 50.00% and specificity of 95.45%. In MB leprosy patients *vs.* ECs with AUC of 0.91 (95% CI [0.81–1.00]), sensitivity of 66.67% and specificity of 95.45%. However, sensitivity was 90.00% in comparison between PB leprosy *vs* ECs and specificity was 81.82%. MCP-1 is more sensitive in PB leprosy diagnosis (sensitivity 100.00% and specificity 66.67%) compared to MB leprosy diagnosis. Both sensitivity (72.22%) and specificity (82.35%) were lower in comparison between leprosy *vs* household controls (HHCs). Overall, MCP-1 is more specific rather than sensitive in diagnosing leprosy, however other study found that MCP-1 are sensitive only to PB leprosy (*Yuan et al., 2021*). Similar findings of higher MCP-1 in household controls (HHC) paucibacillary (PB) as compared to HHC multibacillary (MB) were found in a cross-sectional study conducted by *Cunha et al. (2023)*.

Besides that, MCP-1 was found higher in MB patients in two studies conducted by *Meneses et al. (2014)* and *Queiroz et al. (2021)* (*Dias et al., 2021*; *Queiroz et al., 2021*). Urinary MCP-1 was shown to be greater in multibacillary patients (122.1 ±91.9 *vs.* 72.0 ±46.1 mg/g-Cr, $p = 0.023$) than in paucibacillary patients. Additionally, a significant association was found between urine MCP-1 and the bacteriological index in skin smears ($r = 0.322$, $p = 0.035$). However, MCP-1 levels and the duration of symptoms were not significantly correlated ($r = 0.014$, $p = 0.938$) (*Meneses et al., 2014*). *Queiroz et al. (2021)* found that during the initial visit MB patients had higher levels of MCP-1 than PB patients. However, MCP-1 expression was found higher after 1 year of treatment in PB patients. A significant association ($R2 = 0.05/p = 0.02$), as well as negative correlation ($r = -0,25/p = 0.00$) between MCP-1 and IFN-$\gamma$ was found only in HHC group (*Queiroz et al., 2021*).

MCP-1 can also be used as a predictive value for the occurrence of a reversal reaction (or type 1 reaction). This was stated by a prospective cohort study in 2019 (Tio-Coma et al.) on 10 patients that MCP-1 is useful in comparing the development of reversal reaction (RR) (patients who developed RR ($n = 30$) *vs* did not developed RR ($n = 184$)) because MCP-1 was significantly increased reversal reaction (RR) patients ($p < 0,05$) (*Tió-Coma et*

*al., 2019*). However, GG genotype and G allele at the MCP-1 gene −2518 location confer susceptibility for the emergence of ENL reaction, this finding implies that MCP-1 plays a critical role in controlling immune responses during and ENL reaction (*Biswas et al., 2024*).

Even though most studies we reviewed had shown that MCP-1 was beneficial in diagnostic and predictive outcome, some studies stated that MCP-1 did not have significant diagnostic properties (*Stefani et al., 2009*; *Geluk et al., 2010*; *Mendonça et al., 2010*). A cross-sectional study by *Geluk et al. (2010)* found that production of MCP-1 in response to ML2531 p1-15 and IL-12 tended to be increased by IL-12, although this was not statistically significant ($P = 0.2$ and $0.4$) (*Geluk et al., 2010*). *Stefani et al. (2009)* discovered that MCP-1 levels for non-reactional type 1 reaction-controls (T1R-controls) and type-2 reaction-controls (T2R-controls) groups were not statistically significant (*Stefani et al., 2009*). *Mendonça et al. (2010)* conducted a cross-sectional investigation and found that there were no significant variations in plasma concentrations between infected and non-infected persons among 33 leprosy patients before and during multi-drug therapy (MDT) (*Mendonça et al., 2010*).

## DISCUSSION

This systematic review investigated MCP-1' potential in relation to leprosy diagnosis. Seventeen studies formed the qualitative analysis. Regarding its diagnostic skills, there was a significant degree of variation among the included studies. Some studies were investigating its ability to diagnose leprosy and differentiate between controls; some were investigating the tendency of leprosy' reaction occurrence; some were measuring levels of MCP-1 in different types of leprosy; some were discovering its genetic properties, and some were assessing different levels of MCP-1 between each leprosy classification. Most studies used ELISA to measure MCP-1 levels, some used PCR, and others assessed histopathological staining under the microscope. In general, illustration of role of MCP-1 on leprosy was visualized on Fig. 4.

### MCP-1 as a potential diagnostic marker in leprosy

According to *Medeiros et al. (2015)*, MCP-1 was involved in PNL. In macrophages or Schwann cells present in the majority of nerves with leukocytic inflammatory infiltrate, MCP-1 levels were shown to be higher. This occurred because of Schwann cells' capacity to coordinate a response to peripheral nerve injury, including leprosy nerve damage. Leukemia inhibitory factor release, and IL-6 were released prior to MCP-1 secretion. Following the release of the MCP-1 signal, macrophages began to infiltrate the endoneurial compartment. MCP-1 expression was linked to nerve fibrosis and was detected in PNL Schwann cell biopsy samples. Because macrophages are essential to the inflammatory healing process, they are implicated in the generation of angiogenic and fibrogenic cytokines. MCP-1 increases the production of the pro-$\alpha$1 chain and TGF $\beta$1 in type I collagen. Therefore, MCP-1 was associated with nerve fibrosis (*Medeiros et al., 2015*).

Previous research has revealed that *M. leprae* may enter the lungs, infiltrate pulmonary epithelial cells, and thrive within them. In cells infected with *M. leprae*, MCP-1 was found

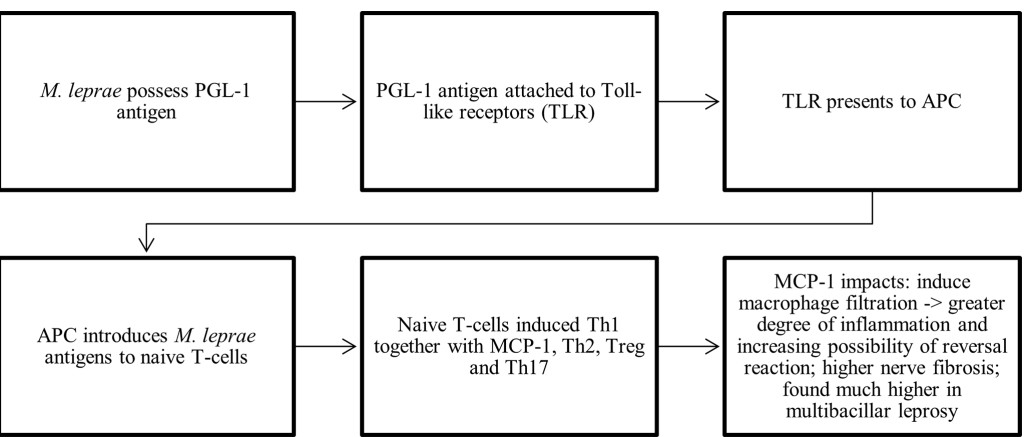

**Figure 4 Schematic findings on the role of MCP-1 in leprosy.** APC, antigen presenting cell; MCP-1, monocyte chemoattractant/chemotactic protein; PGL-1, phenolic glycolipid-1; Th1, T helper 1; Th2, T helper 2; Th17, T helper 17; TLR, toll-like receptor; Treg, regulatory T cells.

to be upregulated. Additionally, exposure to *M. leprae* increased the production of IL-8 in human primary nasal epithelial cells, supporting the possibility that this reaction occurs when the bacteria enter the respiratory system. MCP-1 functions as a chemoattractant for CD4+ T cells and monocytes, whereas IL-8 primarily attracts neutrophils—the initial inflammatory cells that arrive at the infection site to limit the spread of germs (*Dias et al., 2021*).

MCP-1 effectively distinguishes leprosy patients from healthy controls. It has been demonstrated that leprosy patients have significantly higher levels of MCP-1 compared to healthy controls. Detection of MCP-1 can also be used to estimate the magnitude of *M. leprosy* transmission level in healthy controls (*Bobosha et al., 2012*). However, there is no specific test to determine whether exposure to HHC will result in leprosy development (*Cunha et al., 2023*). In asymptomatic people with latent infection, MCP-1 may contribute to the integrity of the granuloma by attracting monocytes, memory T cells, and dendritic cells to areas of tissue damage and infection (*Geluk et al., 2012*; *Meneses et al., 2014*). Therefore, there was a considerable increase in MCP-1 in TT/BT leprosy patients compared to healthy controls (*Geluk et al., 2012*). The MB and LL polar forms of leprosy were reported to have higher urine MCP-1 levels in an investigation by *Meneses et al. (2014)*, despite the absence of clinical renal damage in these leprosy patients. Leprosy patients often experienced renal problems due to inflammation caused by *M. leprae*. Renal inflammation in leprosy patients is believed to be associated with the T helper 2 (TH2) response, which is more pronounced in the lepromatous type of the disease. Although chronic kidney disease may not manifest for a long time, urinary MCP-1 has the potential to be a useful early biomarker for identifying individuals at risk (*Meneses et al., 2014*).

Due to the general chemokine and cytokine profile of the AA genotype, TLR4 rs1927914, and other genetic features are connected to the HHC immune response. Reduced exposure to *M. leprae* (HHC coexisting with PB patients) was associated with higher MCP-1 levels. Administration of BCG vaccine enables CD4+ T cells to recognize *M. leprae*-specific

epitopes, therefore increasing production of the inflammatory mediators, including MCP-1 (*De Carvalho et al., 2017*). Similar claims about single nucleotide polymorphisms in TLR genes increasing leprosy susceptibility by raising the likelihood of developing clinical illness or leprosy reactions were found in earlier investigations. Because of its relevance in subclinical infection, MCP-1, which is related to IFN-$\gamma$, is important to be utilized as a metric for early infection monitoring (*Cunha et al., 2023*). MCP-1 was only expressed on the surface of monocytes; it was not expressed on neutrophils or eosinophils, according to studies by *Yuan et al. (2021)*. Increased MCP-1 levels suggest that it plays a role in leprosy etiology. In addition to TLR4, this study also discovered decreased MCP-1 levels in carriers of the TT genotype (TC and TT) (*Santana et al., 2017*). Leprosy in household contacts is not solely associated with immunological characteristics; other contributing factors include the physical environment of the home, access to latrines, clean water sources, facilities for waste disposal, personal cleanliness, and nutritional condition. Improved hygiene lowers the risk of leprosy among household contacts (*Prakoeswa et al., 2020a*; *Prakoeswa et al., 2020b*).

## Role of MCP-1 in leprosy diagnosis, subtyping, and reversal reaction prediction

MCP-1 is a chemokine ligand that is surface-expressed on monocytes and is implicated in inflammatory reactions and immunological regulation (*Kabala et al., 2020*). Variations in MCP-1 levels throughout leprosy subtypes suggest that this marker can be used to categorize the illness. MCP-1 is typically more specific than sensitive for leprosy diagnosis, especially in PB leprosy (*Yuan et al., 2021*). Lower exposure to *M. leprae* in HHC (PB) was linked to a modulatory axis (marked by greater MCP-1 and IL-10 levels); whereas higher exposure to *M. leprae* in HHC (MB) did not exhibit any modulatory axis. Therefore, it can be utilized as a measurement tool for monitoring early infection in PB patients (*Cunha et al., 2023*). In lepromatous form patient cell cultures, TNF-induced MCP-1 expression was found to be lower, which may have contributed to the dissemination of the bacillus and the development of a more robust inflammatory process in MB patients (*Queiroz et al., 2021*).

This result, however, is incompatible to some other research that discovered elevated MCP-1 levels in MB patients (*Meneses et al., 2014*; *Queiroz et al., 2021*). One possible explanation is that MCP-1 was initially higher in MB patients at the time of diagnosis and later became higher in PB patients one year after treatment. Increased MCP-1 in PB indicated a strong cellular immunological response, which may operate as a leprosy protective factor (*Queiroz et al., 2021*). This statement was supported by a study conducted by *Prakoeswa, Rumondor & Prakoeswa (2022)*, which found that PB patients had higher Th17 cell counts, resulting in better clinical symptoms and a stronger immune response, thereby corroborating this claim (*Prakoeswa, 2022*; *Zaniolo & Damazo, 2023*).

Reversal reactions (RRs) may occur during, prior to, or following MDT. Although previous research suggested that genetic predisposition plays a role in the immunological shift from Th2 to Th1 in RRs, the precise mechanism of RRs remains unclear (*Tió-Coma et al., 2019*). The leprosy reaction is linked to Th1 cells (*Prakoeswa et al., 2020a*; *Prakoeswa et al., 2020b*). Clinical results for RR could be significantly improved by early diagnosis,

particularly in terms of minimizing nerve damage, yet there is currently no established biomarker for RR (*Tió-Coma et al., 2019*). But according to our reviews, future RR patients had higher levels of MCP-1 because of its correlation to excessive extracellular matrix deposition and macrophage recruitment, which triggers pro-inflammatory cytokines and draws CD4+ T cells. This could be because the immune system is exposed to more *M. leprae* antigens following MDT, as indicated by future RR patients' similarly elevated expression of IL-2 (*Tió-Coma et al., 2019*).

While MCP-1 may hold potential for predicting reversal reactions, no statistically significant difference in its levels was observed between type 1 and type 2 reactions. However, GG genotype and G allele at MCP-1 gene 2518 location showed its potential in diagnosing ENL reaction (*Biswas et al., 2024*). *Stefani et al. (2009)* reported a lack of correlations between the duration of response symptoms and the levels of cytokines or chemokines, possibly due to an inadequate sample size. In contrast, *Mendonça et al. (2010)* noted elevated MCP-1 plasma levels in patients with PB; however, it is important to note that all patients in the current investigation were MB, which suggests that the specific MB type may have masked the elevated MCP-1 levels (*Mendonça et al., 2010*; *Gautam et al., 2021*).

**Clinical implications**

These findings imply the possibility that MCP-1 may serve as a diagnostic biomarker for leprosy. Most of our included studies used humans as the study population; however, there were still too few studies for each diagnostic parameter. Future research with larger populations, lower risk of bias, assessments of confounding variables, and systematic procedures for sample retrieval is needed. Several potential areas for future research include studies focusing on MCP-1's diagnostic properties for differentiating between leprosy patients and healthy controls, assessing MCP-1's predictive value in predicting leprosy reactions, distinguishing between different types of leprosy, and identifying genetic properties to predict leprosy's prognostic values. Along with the earlier statement, the variety of the included studies in terms of diagnostic characteristics, different parameters of studies' variables and varying results, it is tricky to reach firm conclusions. To ensure MCP-1's ability to diagnose leprosy and its clinical staging, additional research is needed.

## LIMITATIONS

This study was limited to a systematic review and did not proceed to a meta-analysis due to the heterogeneity of the included studies and because each of these studies assessed different parameters, making meta-analysis impossible to conduct. During the 'Quality and Risk of Bias Assessment' process, we found that most of our included studies did not explain the investigation of potential confounders. None of the case-control and cohort studies stated the ascertainment of exposure. Two out of four cohort studies did not mention the adequacy of follow-up for their cohorts. Additionally, each study acknowledged its limitations. Several studies were excluded because there was insufficient data, and the sample size was lowered due to the non-availability of blood samples. Another potential bias arose from data collection performed by different examiners. The results may also

be affected by using multiple comparisons without correcting for confounding variables and different sample retrieval environments. More studies involving a larger population of leprosy patients and healthy controls are needed to determine which biomarker profiles are best for discriminating *M. leprae*-infected individuals from controls.

## CONCLUSIONS

In conclusion, our systematic review suggests that MCP-1 possesses diagnostic potential for leprosy. The cross-sectional, case-control, and cohort studies included in this review have consistently shown significant associations of MCP-1 levels with the leprosy group, despite the findings of three out of seventeen included studies indicating otherwise. Moreover, MCP-1 has the potential to be beneficial in predicting the occurrence of reversal reactions in leprosy. Therefore, further studies with larger sample sizes and standardized methodologies covering various parameters are necessary to confirm MCP-1's diagnostic properties.

## ACKNOWLEDGEMENTS

The authors acknowledge all the members of Hansen Disease's Research Group(HDRG) of Universitas Muhammadiyah Surakarta for supporting the entire process of this study with conceptualization and peer discussion.

### Funding
The authors received no funding for this work.

### Competing Interests
The authors declare there are no competing interests.

### Author Contributions
- Flora Ramona Sigit Prakoeswa conceived and designed the experiments, performed the experiments, analyzed the data, authored or reviewed drafts of the article, and approved the final draft.
- Ellen Josephine Handoko conceived and designed the experiments, performed the experiments, analyzed the data, prepared figures and/or tables, authored or reviewed drafts of the article, and approved the final draft.
- Erika Diana Risanti conceived and designed the experiments, performed the experiments, analyzed the data, authored or reviewed drafts of the article, and approved the final draft.
- Nabila Haningtyas analyzed the data, prepared figures and/or tables, and approved the final draft.
- Nasrurrofiq Risvana Bayu Pambudi analyzed the data, authored or reviewed drafts of the article, and approved the final draft.
- Muhana Fawwazy Ilyas conceived and designed the experiments, analyzed the data, prepared figures and/or tables, and approved the final draft.

## Data Availability

This is a systematic review/meta-analysis.

## Supplemental Information

Supplemental information for this article can be found online at http://dx.doi.org/10.7717/peerj.17400#supplemental-information.

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
