# Peer review of "The role of monocyte chemoattractant protein-1 (MCP-1) as an immunological marker for patients with leprosy: a systematic literature review"

_PeerJ, doi:10.7717/peerj.17400_

## Round 0.1 · original submission · Major Revisions

Please address the concerns of both reviewers and amend the manuscript accordingly.

**Language Note:** The review process has identified that the English language must be improved. PeerJ can provide language editing services - please contact us at [email protected] for pricing (be sure to provide your manuscript number and title). Alternatively, you should make your own arrangements to improve the language quality and provide details in your response letter. – PeerJ Staff

·

Basic reporting

The manuscript has been well organized. Only a few writing errors were found. Such as inaccurate use of tenses and inconsistent use of abbreviations. Improved English language is needed to ensure that audiences can clearly understand the text. Some examples where improvement can be made include lines 65, 195, 267, 269, 276, 293, 305, 368, 372.

Experimental design

No comment.

Validity of the findings

No comment.

Additional comments

No comment.

Reviewer 2 ·

Basic reporting

In this manuscript, Flora Ramona Sigit Prakoeswa and colleagues perform a literature search, centered around keywords, to explore the role of MCP-1 in leprosy as a diagnostic tool and in predicting reaction occurrences. The subject matter aligns well with the interests of PeerJ readers. Nevertheless, in its present form, the paper does not meet the criteria for publication. A major revision is recommended, as detailed in the comments below.
1. The manuscript requires language improvement to ensure clarity for an international audience. Specific lines where the phrasing makes comprehension difficult include 52, 55, 58, 66, 69, 76, and 80. I recommend having the manuscript reviewed by a colleague proficient in English and familiar with the subject matter, or engaging a professional editing service.
2. The abstract needs to be consolidated to one paragraph that can summarize the introduction, method and results.
3. I did not find the raw data; please ensure that the raw data for the analysis is provided
4. The authors have included an excessive number of self-citations in their references. They need to complement these with additional relevant publications in the field concerning MCP-1 and leprosy. Furthermore, the format of the references needs correction, as exemplified by references 8 and 26.

Experimental design

The literature screening methods employed in this study are not efficient, seemingly overlooking significant research related to the role of MCP-1 in leprosy as a diagnostic tool. The authors have identified only 13 publications for this review, a number insufficient to support a 'systematic review.' They need to revisit and optimize their literature screening methodology or incorporate additional criteria to obtain a sufficient number of relevant publications for a comprehensive review.

Validity of the findings

As mentioned in the 'experimental design' section, the number of literature sources is insufficient to support their conclusions.

Additional comments

1. The authors have repeatedly used titles such as 'MCP-1 diagnostic, genetic, and neuropathic properties' and 'MCP-1 to classify leprosy types and reactions' in the results and discussion sections. I suggest that the authors change these subtitles to clearly indicate the summaries of the content
2. The quality of all three figures and tables needs improvement, as their current quality is not suitable for publication. Additionally, I suggest the authors add a schematic to summarize their findings from the research.

---

## Round 0.2 · accepted · Accept

All issues pointed by the reviewers were addressed and revised manuscript is acceptable now.

·

Basic reporting

No comment.

Experimental design

No comment.

Validity of the findings

No comment.

Additional comments

No comment.

Reviewer 2 ·

Basic reporting

The resubmission addressed all my concerns and I recommend the paper to be published in the current form.

Experimental design

no comment

Validity of the findings

no comment